# Exploring Barriers to the Adoption of Internet of Things-Based Precision Agriculture Practices

Gaganpreet Singh Hundal [1,*], Chad Matthew Laux [2], Dennis Buckmaster [3], Mathias J Sutton [2] and Michael Langemeier [3]

1 Sociotechnical Systems Research Center, Massachusetts Institute of Technology, Cambridge, MA 02139, USA
2 Purdue Polytechnic, Purdue University, West Lafayette, IN 47905, USA
3 College of Agriculture, Purdue University, West Lafayette, IN 47905, USA
* Correspondence: ghundal@mit.edu

**Abstract:** The production of row crops in the Midwestern (Indiana) region of the US has been facing environmental and economic sustainability issues. There has been an increase in trend for the application of fertilizers (nitrogen & phosphorus), farm machinery fuel costs and decreasing labor productivity leading to non-optimized usage of farm inputs. Literature describes how sustainable practices such as profitability (return on investments), operational cost reduction, hazardous waste reduction, delivery performance and overall productivity might be adopted in the context of precision agriculture technologies (variable rate irrigation, variable rate fertilization, cloud-based analytics, and telematics for farm machinery navigation). The literature review describes low adoption of Internet of Things (IoT)-based precision agriculture technologies, such as variable rate fertilizer (39%), variable rate pesticide (8%), variable rate irrigation (4%), cloud-based data analytics (21%) and telematics (10%) amongst Midwestern row crop producers. Barriers to the adoption of IoT-based precision agriculture technologies cited in the literature include cost effectiveness, power requirements, wireless communication range, data latency, data scalability, data storage, data processing and data interoperability. Therefore, this study focused on exploring and understanding decision-making variables related to barriers through three focus group interview sessions conducted with eighteen ($n = 18$) subject matter experts (SME) in IoT-based precision agriculture practices. Dependency relationships described between cost, data latency, data scalability, power consumption, communication range, type of wireless communication and precision agriculture application is one of the main findings. The results might inform precision agriculture practitioners, producers and other stakeholders about variables related to technical and operational barriers for the adoption of IoT-based precision agriculture practices.

**Keywords:** Internet of Things; wireless sensors; precision agriculture practices; barriers

## 1. Introduction

Row crop corn and soybean production in the Midwestern US region has economic and environmental concerns, based on an increasing trends in the application of nitrogen and phosphorus fertilizers [1]. Operational costs and fuel consumption costs have increased with a decrease in labor productivity [1]. As a result, sustainable practices such as profitability (return on investments), operational cost reduction, hazardous waste reduction, delivery performance and overall productivity might be adopted through IoT-based precision agriculture applications, namely variable rate irrigation, variable rate fertilization, cloud-based analytics, and telematics for farm machinery navigation to improve agriculture operations and net profitability. However, [2] note the low rate of adoption for variable rate fertilizer (39%), variable rate pesticide (8%), variable rate irrigation (4%), cloud-based data analytics (21%) and telematics (10%) amongst Midwestern US row crop producers. The technical and operational barriers include operational costs, power consumption requirements, communication range limitations, data latency, data scalability, data storage,

data processing and data interoperability and are highlighted in the literature [3–5]. This paper explores answers to the following questions: what are barriers to the adoption of IoT-based precision agriculture (PA) practices among row crop producers, how are the barriers operationally defined, and how are they related to each other? Section 2 highlights research background, discussing related PA literature on technologies and barriers to the adoption of IoT-based PA practices, Section 3 describes materials and methods adopted for this study based on semi-structured focus group interviews conducted with subject matter expertise (SME) (*n* = 18), Section 4 presents results from thematic content analysis of interview data describing operational definitions of variables and their relationships, Section 5 discusses results and significance in reference to previous studies, and Section 6 describes conclusions highlighting major takeaways, limitations and future research.

## 2. Background

This section highlights research background supported by literature review on PA practices, barriers to the adoption of IoT-based PA practices, current adoption in the Midwestern US region and IoT-based wireless sensor technologies.

### 2.1. Precision Agriculture (PA) Practices

Precision agriculture (PA) practices foster optimized application of agriculture inputs, including seeds, fertilizers, water, pesticides and energy that result in savings on the input applications, resulting in increased yield and improved profitability. Precision agriculture (PA) practices potentially provide producers with improved tools to manage inputs and optimize factors of production such as fertilizer, pesticides and seed application. The definition of precision agriculture published by the National Research Council (1997) defines precision agriculture as "a management strategy that uses information technology to bring data from multiple sources to bear on decisions associated with crop production". Precision agriculture (PA) tools include information-gathering tools such as yield monitors, targeted soil sampling and remote sensing tools; variable rate technology; and guidance systems such as light bars and auto steer equipment. Precision agriculture technologies include soil mapping, variable rate application, yield monitoring mapping, automatic steer global position guidance systems and autonomous vehicles [6]. Management zones in the field are developed by using crop and field information. Varying input rates increase yields or reduce costs depending on the managers' goal for the management zones [7].

The potential benefits of PA practices include an increase in the accurate placement of inputs, reduction of machinery costs from an increase in machinery field capacity and reducing greenhouse gas (GHG) emissions due to reductions in input usage for a given level of production [8,9]. Precision agriculture (PA) practices have an impact on increasing profitability, reducing operational costs, increasing labor productivity, reducing cycle times of operation, optimizing fertilizer (nitrogen & phosphate application) and decreasing fuel consumption in farm machinery [10]. These technologies may consist of variable rate application (water & fertilizer), real-time kinematic (RTK) autosteer, guidance systems (GPS-guided autosteers, yield monitors), submeter accuracy auto steering (SUB) and telematics. In a study by [11], the authors note the impact of PA technologies, such as autosteer guidance, automatic section control spray application and real-time kinematic precision tractor operations on the carbon emission and economic operational cost in the corn and soybean production in the US state of Kentucky. The findings indicate that automatic section control spray application has the capability to spray more precisely, reducing the over-application of inputs and giving a mean net return of 0.47% [11]. Real-time kinematic precision tractor operations provided the most significant improvement in carbon footprint ratio of 2.74% with increased technical efficiency in applying nitrogen and seeds more accurately [11]. Labor productivity also increased, allowing more desirable production practices to be employed [11]. Global navigation satellite systems-based autosteering reduces overlap between tractor passes and overall operator fatigue increasing productivity [12]. A study conducted by [13] to assess the impact of implementation of PA practices in

potato and olive production highlighted that variable rate application (VRT) of potassium and phosphorus fertilizers lead to a strong reduction in nutrient use and an increase in operational profits of 21%, with an increase in overall profits of 26%. Precision agriculture (PA) can help in managing crop production inputs in an environmentally friendly way by utilizing site-specific knowledge targeting rates of fertilizers, seed and chemicals in improving soil conditions [14].

### 2.2. Adoption of Precision Agriculture Practices among Midwestern Row Crop Producers

The Midwestern (Indiana) region of the US has been facing environmental and economic sustainability issues. There has been an increase in trends for the application of fertilizers (nitrogen & phosphorus) and in farm machinery fuel costs and a decrease in labor productivity leading to non-optimized usage of farm inputs [1]. Precision agriculture (PA) technologies foster optimized application of agriculture inputs, including seeds, fertilizers, water, pesticides and energy, resulting in savings in the input applications, resulting in increased yield and improved profitability [12]. The adoption of PA technologies among row crop producers in the US Midwest region has been increasing [2]. The study conducted by [2] notes the adoption of different precision agriculture technologies among Midwest region producers. Yield monitoring technology has the highest adoption rates as 69% of farmers reported adopting it. Variable rate fertilizer (39%), variable rate pesticide (8%), variable rate irrigation (4%), cloud-based data analytics (21%) and telematics (10%) all reported lower adoption rates [15–17]. Different types of IoT sensors (temperature, humidity, light, pressure, wind speed) receive and collect data managed by cloud information management systems for data analysis solutions through application programming interfaces (API) [17]. A wireless sensor IoT framework for PA applications consists of: (1) perception layer, (2) communication layer, (3) processing layer and (4) application layer [4]. Therefore, it is important to understand the barriers and related decision factors involved in developing IoT-based PA applications.

### 2.3. Barriers to Adoption of IoT-Based Precision Agriculture

Multiple studies note socio-economic factors (farmers' educational level, age), agro-ecological factors (soil quality, farm size, ownership of land), farmers' perception (perceived benefits vs. profitability), technological factors (computer education, data aggregation) and informational factors (extension services) as factors having a positive relationship with adoption [16,17]. The research study focuses on cost, technical, operational and data management barriers highlighted in Figure 1 and validated in the study conducted by [3–5,7]. The factors include enumerating cost, power consumption, communication range, data latency, data scalability, data storage and data interoperability [3–5,8]. A more recent study conducted by [5,6,8] highlighted cost of equipment, less benefit, training, data scalability, communication range and time of implementation as the IoT precision agriculture barriers. Figure 1 highlights cost, operational, technical and data management barriers explored in this study.

Figure 2 highlights the different layers of IoT framework [4] for PA practices. This framework was used to develop the semi-structured interview questionnaire highlighted in Appendix B, where questions related to barriers are categorized under these framework layers: (1) perception layer, (2) communication layer, (3) data processing and (4) application layer.

Structured literature review (SLR) methodology described in the following section was used to identify technical and operational barriers to adoption of IoT-based PA practices. Internet of things (IoT) wireless sensors-based PA technologies, namely monitoring row crop diseases, smart irrigation, smart fertilizing, cloud-based analytics and telematics for farm machinery navigation, have a low rate of adoption among Midwestern (Indiana) region row crop producers. These barriers include technical (power limitations, communication range), operational (data scalability, sensor distribution, data latency), management (data storage, data interoperability & data processing) and finance (cost, return on investment).

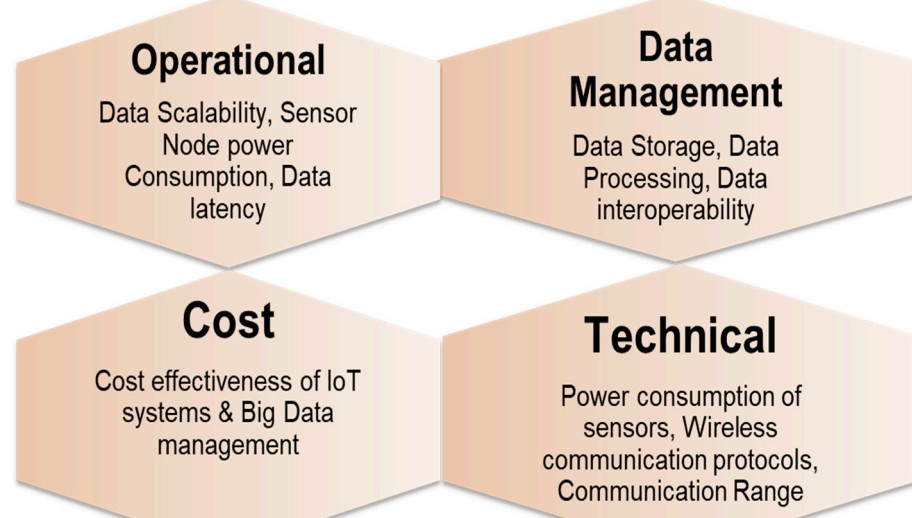

**Figure 1.** Cost, Technical, Operational and Data management barriers. Source: "Reprinted/adapted with permission from Ref. [4]. 2019, Junhu Ruan".

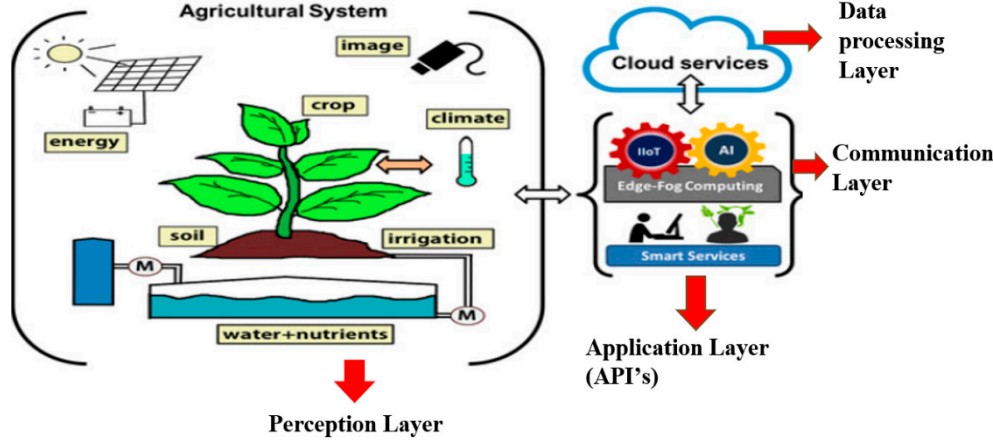

**Figure 2.** IoT framework layers for wireless sensors-based PA. Source: "Reprinted/adapted with permission from Ref. [4]. 2019, Junhu Ruan ".

## 3. Materials and Methods

This section describes the research methodology approaches adopted for this study. Structured literature review conducted for this study is described in Sections 3.1 and 3.2 describes semi-structured focus group interviews conducted with (*n* = 18) subject matter expertise (SME), Section 3.3 describes thematic content analysis approach and Section 3.4 highlights validity and reliability of research methods.

### 3.1. Structured Literature Review

The research methodology framework followed a structured literature review and thematic content analysis of semi-structured focus group interviews conducted with subject matter expertise. Content analysis is defined as "content analysis is a research technique for the objective, systematic and quantitative description of the manifest content of communication" [18]. The four-step process model of content analysis by [19,20] delimits the material to be analyzed by defining a unit of analysis, creating analytical categories, defining the material collection (creating and defining categories), pretesting the categories defined, refining through pretesting categories and analyzing the data by coding for thematic analysis. The first step was defining the unit of analysis, i.e., sample size of relevant peer reviewed

journals and data reports. The period of analysis chosen was 1990–2019, and in the initial review a sample of *n* = 90 journals was selected. The peer-reviewed literature was selected from top-tier publishers Elsevier, Emerald, IEEE, Taylor Francis and Inderscience.

The second step defined screening categories with inclusion and exclusion criteria. Preliminary research criteria were defined based on the broader research objectives, screening the selected unit of analysis, relevant theories, and a review of the initially selected sample journal articles. Internet of Things (IoT)-based precision agriculture technologies were used as initial inclusion or screening criteria. Abstract, introduction and findings of the initial sample were analyzed as per the research objectives, and 20 articles were screened out of 90 articles not meeting the initial screening criteria. The third step was pretesting and refining categories in which the sample of *n* = 70 articles was carefully analyzed for progressive refining and validating of the category scheme. The fourth and the last step was coding and analyzing the themes, as per defined categories in the previous step. The content themes identified for this study were based on keywords from the research questions explored in this research, i.e., "precision agriculture technologies" and "Barriers to adoption of IoT-based precision agriculture practices" were also used to develop the semi-structured focus group interview questionnaire in Appendix B

### 3.2. Focused Group Semi-Structured Interviews

The data collection methods included focus group semi-structured interviews. In this study, a purposive sampling was adopted [21] (p. 94) to collect data from the SME's involved in three focus groups interview sessions. The focus groups were categorized based upon the knowledge and expertise of the participants in three layers of IoT wireless sensors framework: (1) perception layer, (2) communication layer, and (3) data processing & application layer. The credentials of the participants and their current roles and expertise in the IoT framework layer are highlighted in Appendix A. The researcher contacted 22 individuals, of which 18 participated (response rate of 81%) in 3 focus group interview sessions (See Supplementary Material) with 6 participants in each session. Each of the focus groups consisted of at least one expert representing each layer of the IoT wireless sensor framework to reduce bias. The study conducted by [22] defined the term "focus group" to apply to a situation in which the interviewer asks group members very specific questions about a topic after considerable research. Focus groups are used in the studies to investigate complex systems where the research can interact with participants and there is further opportunity to ask for clarification questions. The study by [21] (p. 18) defines a focus group as a "carefully planned discussion designed to obtain perceptions in a defined area of interest in a permissive, non-threatening environment". The critical element of focus group interviews is the involvement of people where the information encourages a nurturing environment [23]. Diagnosing potential problems, programs, services, products and stimulating new ideas results in frameworks for further validation with empirical research [24,25]. Because the characteristic of focus group interviews helps to investigate complex systems where the research may interact with the participants providing an opportunity to further ask for clarification questions, it was appropriate to adopt these research methods [21] (p. 18). The studies conducted by [23,24] highlighted that purposive sampling may be employed for focus group interviews that consist of representative members of larger populations. Most focus groups interviews consist of between 6–12 participants [25]. The study conducted by [25] suggests that "the size of the group is governed by the objectives of the research as well." Smaller groups (4–6 people) are preferable when participants share information about the topic [21] (p. 24). The study also highlighted that the typical focus group interview might have up to 10–15 questions, depending upon the length of the interview and research objectives. The semi-structured interview questionnaire in Appendix B, developed for this study, consisted of 12 questions structured in four sections, namely (1) demographics (2) perception layer (3) communication layer and (4) data processing & application layer.

### 3.3. Thematic Content Analysis

Data analysis is the part of qualitative research that mostly distinctively differentiates from quantitative research methods [26]. Qualitative data analysis is a more dynamic, intuitive and creative process of inductive reasoning, thinking and theorizing [27]. Data analysis in qualitative research is defined as the process of systematically searching and arranging the interview transcripts. The process of analyzing qualitative data involves coding or categorizing of data, which reduces the volume of raw information, followed by identifying significant patterns drawing meaning from the data, developing a logical chain of framework or adding to grounded theory [28]. The NVivo coding software was used to code the data from interviews. Inter-reliability of the coding thematic analysis of the data was checked with another rater to ensure the reliability kappa value k > 0.70 for acceptable reliability [29,30].

### 3.4. Validity and Reliability

Validity and reliability of the research design for this study follows a triangulation method, which is a typical practice for rigor in qualitative research methodologies [29,30]. Triangulation may involve combining multiple data sources (data triangulation), using multiple research methods to analyze the same problem (methodological triangulation) or using multiple investigators [31]. In this study, the approach of triangulation was followed by the researcher collecting and analyzing data from different sources, i.e., the SLR (*n* = 70) and the three focus group interview sessions. The content analysis from the focus group interviews was triangulated by three non-subject matter expertise appraisers for inter-reliability.

## 4. Results

This section highlights the results and findings from thematic content analysis conducted on the focused group interviews (See Supplementary Material). The frequency of emerging themes from content analysis conducted on focus group interviews is shown in Figure 1. How the variables are defined in the context of barriers and their potential relationships with each other are described in Table 1. Subsequently operational definitions for variables identified and defined through content analysis are highlighted in Table 2.

**Table 1.** Variable Relationships and Descriptive Content Analysis.

| Variable Relationship | Participant Response | Avg. Kappa Value (Inter-Reliability) | Descriptive Analysis |
|---|---|---|---|
| Data interoperability-Data storage | P2: "You know interoperability of formats and everything, something that will happen as add data gets larger and you might want to go between different cloud environments, so we have some technology that we are developing to do some of that not entirely pivoted to other, but multi cloud is for example" P3: "Data interoperability it's more at the software like application level. Different software companies may need to talk to each other to make sure the results generated by them can be used by each other." | 0.33 (Weak agreement) | Data storage tends to depend upon the data interoperability requirement. The interoperability between sensors, wireless communication technologies and cloud storage-end. The requirements for storage less or more tend to depend upon the compatibility of storage (sensors, wireless communication gateway, cloud) with each other and precision agriculture application requirements. |
| Data interoperability-Type of sensors | P4: "If I install sensor A and now, I'm stuck with this product and I can't use this one over here because they just don't talk to each other. I mean like if we're really going to get people to adopt, there needs to be choice and it needs to be some flexibility." P5: "Data interoperability is important to find it's a very hard thing to define. You've seen things like at Gateway. Try and maybe they went too far and got really specific in the weeds. There's probably some middle ground. You know that that needs to be the first step. It's like how do we identify the 90% most important data and just come up with formats for that?" | 0.44 (Fair Agreement) | Data interoperability tends to depend upon the type of sensors as interoperability might mean data coming from different types of sources. Participants reported that there will be more data interoperability between the same category of sensors as they get integrated well with a particular type of wireless communication protocol (LoRA, Zigbee, Sigfox, BLE, Wi-Fi, GPRS 3G/4G). |

**Table 1.** *Cont.*

| Variable Relationship | Participant Response | Avg. Kappa Value (Inter-Reliability) | Descriptive Analysis |
|---|---|---|---|
| Data interoperability- Type of Wireless communication | P2: "I think open source obviously has its own benefits. There is the NB- IoT is definitely nice. You can have NB IoT tier towers, but it is more expensive. I think I know that they are connected by doing Lora-WAN connectivity which will be nice. It will have this open-source thing in addition" P5: "As long as this server has the ability to talk to the sensor using the correct wireless communication technology, it doesn't matter much like which technology you are using. Just establish the link and the data flows along the link so in that sense the communication is like a very low level, almost like you don't need to worry about it as long as it's there. For data interoperability it's more at the software like application level." | 0.67 (Good Agreement) | Type of wireless communication technology tends to not depend upon the data interoperability and vice versa. The participants highlighted that data interoperability is more adaptable at the user interface, i.e., software or cloud storage end. |
| Data interoperability-Type of Precision application | P6: "Annotated data set so that you can exactly find out what is the disease and you can use that annotated data set into training your different machine learning or deep learning model. So that is one of the missing pieces, because although we reached out to Plant Village plant dog datasets, but then we had to do the annotation by ourselves. So, a very large-scale annotated data set is still needed and that is one area" P7: "Flow rate, pressure, pH of that so that it is very hard to join that piece of information. The Fertilizing team for the different sections of the farm. So as previously mentioned, integrating like different heterogeneous sensors." | 0.67 (Good Agreement) | Data interoperability tends to depend upon the type of precision agriculture application, type of sensors and cloud-data storage user interface. For instance, as the participants mentioned for monitoring row crop diseases, application of open-source field topography, soil and satellite data apart from the sensors might be used easily to develop robust row- crop disease models. |
| Data Latency- Autonomous applications | P4: "Depending on how latency sensitive that specific activity is, and in general I don't think the latency thing is ever at the millisecond level or the second level. It's always at a higher level of granularity because you know if you're like sensor monitoring, for example soil monitoring. The comparison that I'm making compared to self-driving cars. I guess some of the latency sensitive mapping from the self-driving car industry comes in where you might have to do on device analytics to take care of that latency issue." P3: "Drone that is sensing and at the same time spring then I think it becomes important to at that sub millisecond level that you would actually have to do the computation." | 0.75 (Excellent Agreement) | Autonomous applications tend to have low latency requirements in miliseconds or seconds specifically mentioned by participants for farm machinery navigation systems using GPS and accelerometer sensors. The smart irrigation and smart fertilization where the data from soil moisture, Ph and Nitrate sensors tends to have data latencies requirements in minutes or hours. The monitoring applications specifically for row crop diseases where data latency requirements may be in days or weeks. |
| Data latency- Data scalability | P7: "If there is remote sensing or otherwise you know yes there is time to move that to the right place. Do the computation, generate the prescription, and then send it out. But, also increasingly there are machines that on the front of the machine they sense what needs to happen here and on the back of the machine it happens so in that case there really is no latency like it's. You know it's gotta be within a fraction of a second depending on the speed of the vehicle, certainly anything that's navigation related if it's autonomous, has to be sub millisecond." P8: "So, for autonomous applications one of if you're thinking of autonomous driving in terms of tractors or whatever, you can have a lot of data and you can offline train the model right. So, if you can train the model offline using various kinds of temporal datasets that have been taken overtime, it's going to just enable decisions to be taken at real time faster. Currently it is right you don't want it to make bad decisions, especially for things related to autonomous driving, so I think it's latency sensitive and plus the cost of a bad decision is high." | 0.67 (Good Agreement) | Data latency tends to not depend upon the data scalability requirements as it depends upon the type of precision agriculture application requirements. |

**Table 1.** *Cont.*

| Variable Relationship | Participant Response | Avg. Kappa Value (Inter-Reliability) | Descriptive Analysis |
|---|---|---|---|
| Data latency-Power consumption | P3: "Lora-WAN maybe more specific would be appropriate for like one of these, you know, hundreds or thousands of sensors spread out over a huge area. You know something like that would typically be power or battery, small battery or energy harvested kind of thing. The sensors, probably by nature, not a very fast update rate, or don't measure very often. And so, it maps to that technology well, but LoRA won't solve the problem in the space, one would be just machine automation. You'll never be able to have a cloud connected machine that's maybe utilizing the cloud's ability to do real time computations and have that connected through LoRA like this is probably just never work for latency reasons." <br> P5: "LoRA gateways consume huge amounts of power so that their clients don't have to. You know. I mean, it's a balance. You can only get so much data latency for a certain amount of power." | 0.67 (Good Agreement) | Data latency tends to depend upon the power consumption requirements as lower the latency requirements from sensors and wireless communication technologies higher the power consumption. LoRA is a low power and higher latency wireless communication technology. The 3G/4G/5G and wireless WIFI might have more power consumption as they have low latencies-high data transfer rate and are used for autonomous applications. Bluetooth wireless BLE is a low power- low latency option for shorter communication range precision applications such as for RFID, GPS and other short communication range sensor precision applications. |
| Type of Wireless Communication- Power consumption | P14: But for power we did a comparison for some of the different wireless technologies. We were interested in the power consumption for data transfer and our goal was to decrease the data transfer for it's obviously much lower. Especially Bluetooth low energy, right. So, 85.8% for LoRA it is 99.9% for LoRA and SigFox. So, I think just to map to a higher granularity, BLE is quite a bit lower. So, you may have to consider that if you're thinking of data transfer using these different wireless technologies" <br> P18: "The latency requirements and the power consumption. So, we actually have found that if the latency requirements are low and you basically want to consume the power, and that's where we did this comparison between the different wireless technologies that LoRA is much more power consuming than Bluetooth. Depending on what networking modality you will have and the battery requirements of that sensor, you might want to kind of bound the amount of data transfer that is happening from the sensor to the gateway." | 0.67 (Good agreement) | Type of wireless communication technology tends to depend upon the power consumption requirements as lower latency requirements of the precision application such as autonomous tends to have higher throughput rate and higher power consumption. |
| Data Storage-Power consumption | P4: "Here's the rest API that you can pull the data in from so we can't quantify what the power consumption is for that. But then you have it on the plant side. Sometimes you might have sensors where you could determine how often you want this data. So, then that determines the power consumption that's going to be drawing. And so that's to give like different perspectives on sometimes you might not know how it's affecting the power consumptions. And sometimes you can see immediately that the battery level is one of the sensors that you were looking at and how the power is being drained." | 1 (Excellent agreement) | Data storage tends not to depend upon the power consumption requirements. Power consumption tends to depend upon the data latency requirements as mentioned by participants. |
| Data Scalability-Cost | P7: "If you get, you know that data all the time and then you know if you want to record with ISO-Blue. If you want your real data at the end of the year, you know you don't need that. By the 2nd, get that at the end of the year, you know that the high bandwidth data you don't necessarily need you know." | 0.70 (Good Agreement) | Data scalability tends to depend upon the cost requirements as data from different types of sensors might require more storage capacity that involve costs. However, the increase in storage at the cloud-end or wireless communication gateway might not contribute significantly to cost increase. Deploying large scale and different types of sensors might contribute significantly to the input variable cost. |

**Table 2.** Operational Definitions.

| Variable (Themes) | Supporting Participant Quotes | Definition's Interpretations-Content Analysis |
|---|---|---|
| Cost | P11: "You have to go out there and take subsamples and that isn't near detailed enough to address the variability that's in the field. And so, to me, the big game changer that we really need is some kind of a low-cost accurate phosphorus potassium soil Ph type of a sensor". P2: "The coverage using wireless technologies and for soil sensors LoRA will be great. For typical applications, Wi-Fi will be cheaper. It's like it can support higher throughput; cellular is very promising, but it can cost a lot". | Fixed cost (sensor cost, communication gateway technology cost). Variable cost (sensor batteries cost, power consumption cost, cloud storage subscription cost). |
| Types of Sensors | P14: "So, if you're talking about a piece of autonomous farm equipment, at least of the data latency requirements, they are going to be significantly higher. Anything that involves safety typically requires high data rates and often has redundancies." | Monitoring row crop diseases (weather sensors (temperature, humidity, light, pressure, soil moisture), remote sensing (drones, GPS, LiDAR image sensors), smart fertilization (Ph, nitrate soil sensors), smart irrigation (soil moisture, Ph level), farm machinery autonomous operations (GPS, accelerometers, proximity, fuel level, sound). |
| Type of Wireless Communication | P18: "The latency requirements and the power consumption, we actually have found that if the latency requirements are low and you basically want to consume the power, and that's where we did this comparison between the different wireless technologies that LoRA is much more power consuming than Bluetooth. | 3G/4G/5G, LoRaWAN, Sigfox, TVWS (long communication range > 5–10 miles), mid-range (<1 mile) Zigbee, Wi-Fi, short range (10–1000 m), Bluetooth (BLE), GPRS. |
| Type of Precision Agriculture application | P10: "Autonomous areas don't really care about data that happened previously and you're not really going to need to store historical data, with the exception of the monitoring applications So I don't imagine there being a very high data storage requirement." | Monitoring applications (row crop diseases), autonomous applications (smart irrigation, smart fertilization, farm machinery navigation autonomous operations. |
| Data Storage | P11:"Keeping terabytes of data costs next to nothing on the cloud side or on the edge side. We have 500 gigabyte SSD's that doesn't cost too much and is smaller than a credit card, so that's more than enough for us". | Data storage platforms consists of wireless communication gateway-end and cloud storage (user-end application interface). |
| Data Scalability | P16: "Something on the leaves of crops, then you would use satellite aerial or drone typically, and so, I mean these are sensors or few in number, but they're collecting a lot of data over a wide area." | The scale or amount of data transferred for storage and processing from different types (number) of sensors to edge (communication gateway-end) or cloud (user-end application interface) for developing precision agriculture applications. |
| Data Latency | P14: "If your variable doesn't change too often then it's overkill to be sampling 8 every second so you can't save a lot and probably this is one of the biggest advantages of LoRa. You can cover a big range and send a very low rate." | Refers to the data transfer rate (bits/secs, bytes/secs, kbps, mbps) requirements defined for sensors and wireless communication technologies integrated with sensors for transferring data. |
| Communication Range | P18: "Something like LoRa-WAN maybe more appropriate for hundreds or thousands of sensors spread out over a huge area. Something like that would typically be a small battery or energy harvested kind of thing. The sensors, by nature, not a very fast data transfer rate required, or don't measure very often." | Defined as the communication distance between the sensor node (the device integrated with different types of sensors) and wireless communication gateway technology. |
| Data processing | P10: "So, if you see noisier data, you can do more dimensionality reduction or noise removal. So that is one way of doing differential analytics. The other is related to latency, so if you're concerned about latency you might want to do more of the processing on the device or on the edge or in the cloud depending on the latency requirements." | Defined as the amount of data (bits/secs, bytes/secs, kbps, mbps) processed by the communication gateway technology and cloud storage-end. |

**Table 2.** *Cont.*

| Variable (Themes) | Supporting Participant Quotes | Definition's Interpretations-Content Analysis |
|---|---|---|
| Power consumption | P4: "Sometimes you might have sensors where you could determine how often you want this data. So, then that determines the power consumption that's going to be drawing". P5: "LoRA gateways consume huge amounts of power so that their clients don't have to. I mean, it's a balance and you can only get so much data latency for a certain amount of power." | Defined as power consumption by sensors, wireless communications technology integrated with sensors along with backhaul networks (Wi-Fi, GPRS, 3G/4G/5G). The operating battery voltage for sensor nodes having different types of integrated sensors is the indicator for power consumption used in the empirical analysis for this research. |
| Data interoperability | P2: "You know interoperability of formats and everything, something that will happen as data gets larger and you might want to go between different cloud environments, so we have some technology that we are developing to do some of that not entirely pivoted to other, but multi cloud is for example." | Defined as the ability of cloud storage (user-end application interface) to store and process data from different sources (different types of sensors, wireless communication mobile gateway edge, remote sensing, other open-source data) and communicate well with heterogeneous sensors and farm machinery for data transfer. |

### 4.1. Themes from Content Analysis

Figure 3 highlights the reference frequency of emerging themes identified and analyzed following the content analysis approach on the focus group interviews (See Supplementary Material). The themes analyzed were as per the research questions based on identifying, exploring and defining variable relationships related to cost, technical (power consumption, wireless communication, communication range), operational (data scalability, data latency) and data management (data storage, data processing, data interoperability) barriers.

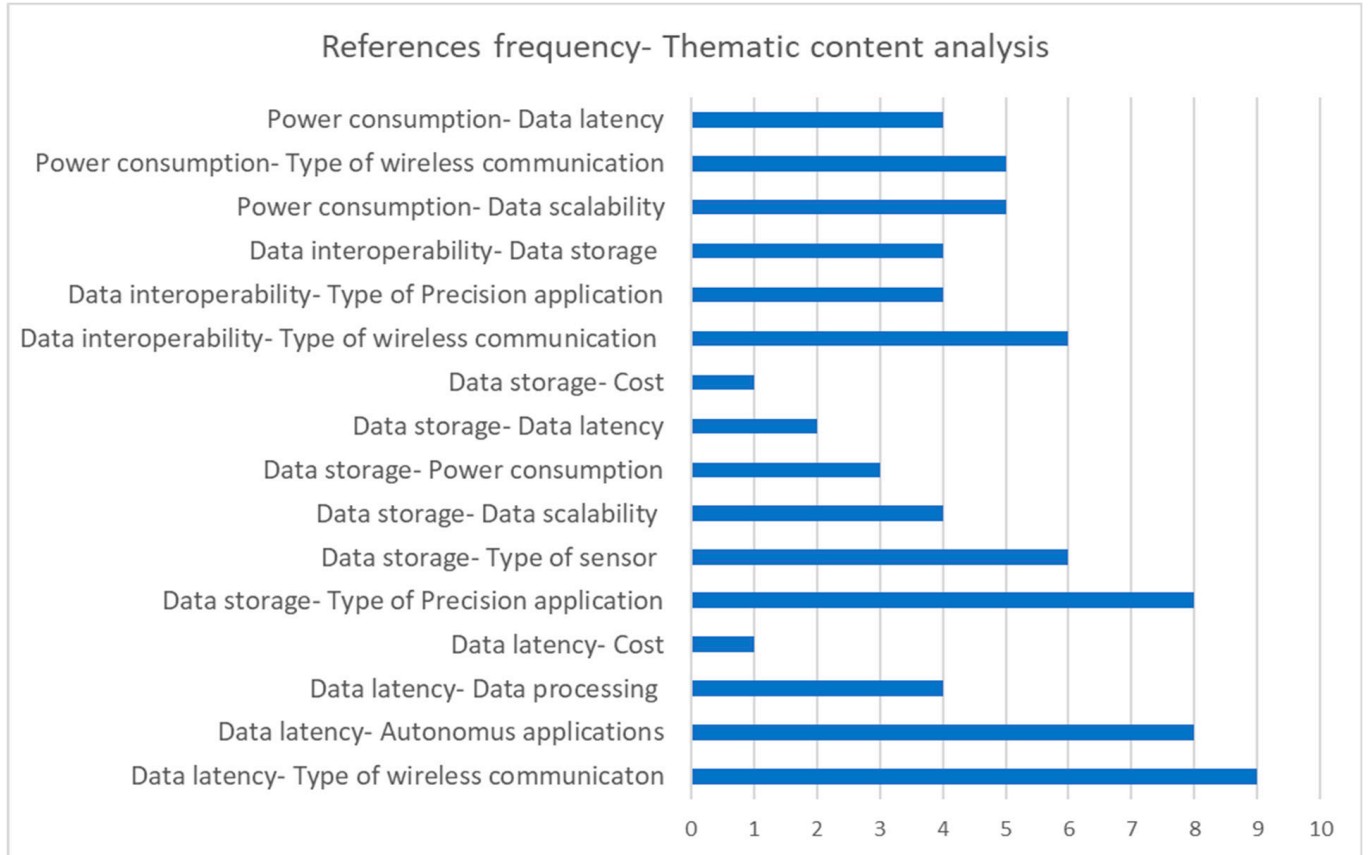

**Figure 3.** Frequency of text references identified in focus group interview transcripts.

The emerging themes identified through content analysis in decreasing order of their reference frequency are as follows. Data latency—type of wireless communication (9), data latency—autonomous applications (8), data storage—type of precision application (8), data interoperability—type of wireless communication (6) and data storage—type of sensor (6) were the most recurring themes identified. Data latency—power consumption (4), data interoperability—data storage (4), data interoperability—type of precision application (4) and data latency—data processing (4) were the other recurring themes identified. The emerging themes were explored further for descriptive analysis, and findings are reported in the following section.

*4.2. Variable Relationships*

The variable relationships identified as themes through the content analysis approach were further explored. The participants' quotations in the context of variable relationships are reported in Table 1, along with descriptive analysis findings. The Kappa value for each of the themes analyzed through content analysis shows the reliability of findings. Cohen's Kappa coefficient is a statistical measure of inter-rater reliability, which many researchers regard as more useful than the percentage agreement since it considers the amount of agreement that could be expected to occur through chance [30]. NVivo calculates the Kappa coefficient individually for each combination of emerging themes [31]. If the two users are in complete agreement about which content of the source should be coded at the node, then the Kappa coefficient is 1. A value between 0 and 1 indicates the scale of agreement. The Kappa (k) value $\leq 0.40$ means weak agreement, between 0.40–0.75 means good agreement and above 0.75 means excellent agreement [31]. The average Kappa (k) value is calculated for each individual node by using the mean k value calculated for each comparison rating between individual raters ($n = 3$ raters) involved, as per the typical practice mentioned in the studies [30,31].

The descriptive analyses for the variable relationships are interpretations from the participants' quotes. For instance, data latency requirements tend to depend upon power consumption, type of wireless communication technology and type of precision application, which might be important to consider before designing an IoT-based data pipeline for precision agriculture applications. Similarly, data scalability depends upon power consumption and cost requirements, which might be critical to understand for cost- and energy-efficient IoT wireless sensors-based data pipeline design.

*4.3. Operational Definitions*

The operational definitions of variables identified through the content analysis approach are highlighted in the table. The operational definitions are stated as findings of content analysis from focus group interviews. The supporting participant quotes are also mentioned in Table 2. Therefore, the subjective interpretations for operational definitions are limited in the context of this study.

**5. Discussion**

The variable relationships in the contexts of technical and operational barriers identified through this research might help to inform precision agriculture practitioners, researchers and producers. The findings are based on potential dependency relationships between cost, data latency, data scalability, data storage, data interoperability, type of sensors, type of wireless communication, type of precision agriculture application and power consumption, which might help to understand barriers from an IoT systems perspective. The findings suggest that data latency requirements depend on the type of precision agriculture application. For autonomous applications, specifically farm machinery navigation, the latency requirements are lower (ms or s) as compared to monitoring applications (days or weeks). Data latency further depends upon power consumption as low latency means higher power consumption. This is an important finding as it suggests that Wi-Fi/5G/4G are the suitable wireless communication technologies for low-latency autonomous appli-

cations. This finding is supported by the studies [3,4,15] where a comparative analysis based on power consumption and communication range is conducted for different types of wireless communication technologies. Another key finding based on data interoperability depends upon the data storage requirements as data collected from different types of sensors require more cloud storage compatibility to interoperate and perform data processing for different types and structure of data [32–34]. This finding resonates with the studies conducted by [4,32,33] which highlights the compatibility of cloud data storage platforms to interoperate for different types and structures of data coming from different types of sensors [32–34]. In addition, the findings of the study suggests that monitoring row crop diseases applications require data from multiple type of sensors (soil moisture, temperature, humidity, topography, satellite imagery data) which require more data interoperable cloud storage platforms [35,36]. The operational definitions for variables defined in Table 2, through descriptive analysis of interview data, are significant results to help digital agriculture practitioners, producers and researchers to understand the adoption barriers. Understanding the variable relationships highlighted in Table 1 might help to design cost- and energy-efficient solutions for IoT-based precision agriculture applications.

## 6. Conclusions

The dependency of data latency requirements on power consumption, type of wireless communication and precision agriculture applications is an important finding to consider for designing IoT-based wireless sensors autonomous applications (i.e., smart irrigation, smart fertilization and farm machinery navigation) [34–36]. Low data latency requirements for autonomous applications and comparatively higher data latency for monitoring applications is an important takeaway which might help to design energy- and cost-efficient IoT systems [32,33,35]. Higher data latency means low power consumption, and therefore using low-power wider-coverage technology such as Lora WAN holds significance. Data interoperability depends upon compatibility of cloud storage to store and process different formats and structures of data collected from different types of sensors [36,37].

Data scalability depends on cost as large data require more storage and processing capabilities at the cloud end; the cost might be insignificant but the cost of more sensors collecting data is significant [37,38]. Further dependency of types of wireless communication on data latency, communication range and power consumption are another critical finding to inform stakeholders for cost- and energy-efficient wireless communication protocol technologies [32,34,35]. The decision variables identified and operationally defined through this study might be explored further for dependency relationships and to develop an empirically validated framework for sustainable IoT. Interpretation of research findings following qualitative content analysis approaches are subjective and have limitations in context generalizability. However, researchers have tried to reduce biases, keeping high rigor for reliability and validity of findings. The findings of this research define and describe potential relationships between variables which might help digital agriculture practitioners and researchers to develop cost- and energy-efficient IoT systems for adoption of precision agriculture practices.

**Supplementary Materials:** The following supporting information can be downloaded at: https://www.mdpi.com/article/10.3390/agriculture13010163/s1, Focused group interview session 1; Focused group interview session 2; Focused group interview session 3.

**Author Contributions:** Conceptualization, G.S.H. and C.M.L.; methodology, G.S.H. & C.M.L.; software, G.S.H.; validation, C.M.L., D.B. & M.J.S.; formal analysis, G.S.H.; investigation, G.S.H., C.M.L. & D.B.; resources, C.M.L., D.B. & M.J.S.; data curation, G.S.H.; writing—G.S.H.; writing—review and editing, C.M.L., D.B., M.L.; visualization, G.S.H. & M.J.S.; supervision, C.M.L., D.B., M.J.S. & M.L.; project administration, C.M.L. & D.B.; funding acquisition, D.B. All authors have read and agreed to the published version of the manuscript.

**Funding:** This research was funded by Wabash Heartland Innovation Network IRB-2020-1819.

**Institutional Review Board Statement:** The study was conducted in accordance with the Declaration of Helsinki and approved by the Institutional Review Board (or Ethics Committee) of Purdue University (protocol code 1819 and 4 January 2021) IRB-2020-1819.

**Data Availability Statement:** The data presented in this study are available on request from the corresponding author. The data are not publicly available due to confidentiality and privacy reasons. However, the request for access can be made using the following link: https://drive.google.com/drive/folders/11DFWtBlYQfwTR472UpWljBhxtxk-mIDa?usp=sharing (accessed on 15 September 2022).

**Acknowledgments:** I would like to thank all the subject matter expertise from Open Agricultural Technology & Systems (OATS) research center at Purdue College of Agriculture, Purdue Agronomy Center for Research & Education (ACRE) and Microsoft Farmbeats community.

**Conflicts of Interest:** The authors declare no conflict of interest.

## Appendix A. Participants Current Role and Expertise Area

| Participants | Current Role | Expertise in IoT Wireless Sensor Framework Layer |
|---|---|---|
| Participant 1 (P1) | Digital Agriculture Technology Consultant | Wireless Communication technologies (Communication Layer) |
| Participant 2 (P2) | Program Coordinator in Agriculture Technology | Big Data Telematics, Data Analytics, Aerial Imagery (Perception Layer) |
| Participant 3 (P3) | Global Technology consultant | Wireless Communication technologies (Communication Layer) |
| Participant 4 (P4) | Graduate Research Assistant | UAV-aided wireless communication systems, Intelligent transportation system applications in Digital agriculture (Communication Layer) |
| Participant 5 (P5) | Precision agriculture technologies consultant and Farm-owner | Digital agriculture practitioner, Smart irrigation & Autonomous precision agriculture application (Application Layer) |
| Participant 6 (P6) | Academic Faculty | Wireless Communication Technologies for Agriculture, Signal processing, Sensor network design (Communication Layer) |
| Participant 7 (P7) | Cloud technologies consultant | Cloud computing platforms for Digital Agriculture (Data processing Layer) |
| Participant 8 (P8) | Graduate Research Assistant | Wireless Communication technologies, Embedded systems & edge-computing (Communication Layer) |
| Participant 9 (P9) | Academic Faculty | Decision Support System, Cloud Computing, Mobile Apps (Application Layer) |
| Participant 10 (P10) | Graduate Research Assistant | Autonomous precision agriculture applications (Application Layer) |
| Participant 11 (P11) | Digital Agriculture Consultant | Internet of Things (IoT) for Farm machinery autonomous applications (Application Layer) |
| Participant 12 (P12) | Graduate Research Assistant | Software engineering, API for crop monitoring applications (Data processing Layer) |
| Participant 13 (P13) | Wireless communications Technology consultant | Wireless communication networking, long range and wide area networks (LoRa) for digital agriculture applications (Communication Layer) |
| Participant 14 (P14) | Graduate Research Assistant | Internet of Things sensors applications for precision agriculture (Perception Layer) |
| Participant 15 (P15) | Extension program coordinator | Digital Agriculture practitioner, Rural area sensor networking (Perception Layer) |
| Participant 16 (P15) | Digital Agriculture practitioner & Farmer | Digital agriculture technologies adoption and practitioner (Perception Layer) |
| Participant 17 (P17) | Software engineer | Cloud computing, Big Data Analytics for IoT in Agriculture (Data processing Layer) |
| Participant 18 (P18) | Application Programming Interface (API) developer | Software developer for Precision agriculture applications, Cloud-back end (Data Processing layer) |

## Appendix B. Interview Script

Hello, my name is _____________________ and I am a Purdue University graduate student conducting this focused group interview with _______________ on (date/time).

The purpose of this research study is to explore the lean (cost, power, data scalability, data processing and user-experience) and green (energy and hazardous waste reduction precision agriculture applications) Internet of things (IoT) wireless sensors framework for the adoption of precision agriculture applications (monitoring row crop diseases, smart irrigation, smart fertilizing, and farm-machinery efficient navigation) amongst row crop

producers in Indiana region. A content analysis will be conducted through focused groups semi-structured interviews with subject matter experts in open-agriculture technological systems (OATS), Digital agriculture experts (Professors, Graduate Students & Purdue ACRE extension members). The findings of the content analysis from the focused group interviews will be used to inform the multiple Farm beats sensor boxes deployment at Purdue Agronomy Center for Research and Education (ACRE) farm facility following action research. The goals of this study are as follows:

Identify the different types of sensor combinations that can be used to gather the data for developing precision agriculture applications (monitoring row crop diseases, smart irrigation, smart fertilizing, and farm-machinery efficient navigation) for an average size row crop farm in the Indiana region.

- Understand the efficient (cost, power, data scalability, data management) and effective (communication range, data latency, data interoperability, data processing) wireless communication technologies that can be integrated with sensors for developing precision agriculture applications (monitoring row crop diseases, smart irrigation, smart fertilizing, and farm-machinery efficient navigation) for an average size row crop farm in Indiana region.
- Understand and identify the efficient (cost, power, data management) and effective (data latency, data interoperability and data management) data storage and processing application programming interfaces for developing precision agriculture applications (monitoring row crop diseases, smart irrigation, smart fertilizing and farm-machinery efficient navigation) for an average size row crop farm in Indiana region.
- Understanding the dependencies of variables namely type of sensors, type of wireless communication technologies, no. of devices-data scalability, communication range, data latency, data interoperability of application programming interfaces with cost, power consumption and type of precision agriculture applications (monitoring & autonomous).

To participate in this research, we ask for approximately (1 h) of your time through a guided semi-structured focused group interview. All research carries risks, but the risks associated with this study are minimal and no more than found day to day. The minimal foreseeable risks are that your identity might be accidentally revealed to parties other than the researchers, should there be a confidentiality breach. However, we are taking several measures to protect your identity. There are few direct benefits to you from participating in this research, but the research results will benefit: row crop farmers, digital agriculture practitioners and open agricultural technology researchers. This interview will be recorded for transcription as data collection for subsequent analysis. We do appreciate your time as your experience, background & expertise are critical to the success of this study.

Do you all consent to participating in this study? May we record our conversation?

**Demographics**

1. What is your educational background and current role in the organization you work?
2. What is your experience with digital agriculture?

**Perception Layer (Types of Sensors)**

1. What are the different types of sensors that can be used to develop applications for monitoring row crop diseases on an average size farm in the Indiana region?
2. What are the different types of sensors that can be used to develop applications for smart irrigation applications on an average size row crop farm in the Indiana region?
3. What are the different types of sensors that can be used to develop applications for smart fertilizing on an average size row crop farm in the Indiana region?
4. What are the different types of sensors that can be mounted on farm- machinery for efficient navigation during planting and harvesting operations?

**Communication Layer (Wireless Communication Technologies)**

1. How can we efficiently (cost, power, scalability) and effectively (communication range, data latency, data storage and processing) integrate different types of sensors with wireless communication technologies for monitoring row crop diseases precision application on an average size farm in the Indiana region?
2. How can we efficiently (cost, power, scalability) and effectively (communication range, data latency, data storage and processing) integrate different types of sensors with wireless communication technologies for smart irrigation autonomous application on an average size row crop farm in the Indiana region?
3. How can we efficiently (cost, power, scalability) and effectively (communication range, data latency, data storage and processing) integrate different types of sensors with wireless communication technologies for smart fertilizing autonomous application on an average size row crop farm in the Indiana region?
4. How can we efficiently (cost, power, scalability) and effectively (communication range, data latency, data storage and processing) integrate different types of sensors with wireless communication technologies for farm- machinery efficient navigation on an average size row crop farm in the Indiana region?

**Data Processing & Application Layer (Data Storage, Management and Processing Applications)**

1. How can we (cost, power, scalability) and effectively (communication range, data latency, data interoperability) store and process data for developing monitoring of row crop diseases and precision application interfaces?
2. How can we (cost, power, scalability) and effectively (communication range, data latency, data interoperability) store and process data for developing autonomous (smart irrigation, smart fertilization & Farm machinery navigation) alert application interfaces?

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
