# Peer review of "Exploring Barriers to the Adoption of Internet of Things-Based Precision Agriculture Practices"

_agriculture, doi:10.3390/agriculture13010163_

Round 1
Reviewer 1 Report
Reviewer’s Comments
The manuscript " Exploring barriers to the adoption of Internet of Things wireless sensor-based Precision Agriculture practices," with the Manuscript id: agriculture-2013215, is well written. Although I recommend accepting the manuscript, I do have a few comments and suggestions that should be made before it is officially accepted.
following are the comments:
1. Author must check for unnecessary capitalization and try to avoid it.
2. Section 2.3: the citation must be continuous (numbers), how just after 18, it appears 35,37, and 39. Need careful citation arrangement.
3. Section 3.1: last sentence of the first paragraph, “The peer ….. Science direct.”. Elsevier and science direct are distinct?
4. Section 3.2: first paragraph on page 6, the citation style must be uniform throughout the manuscript. As the author changed here to “(Stewart & Shamdasani, 1990)”, fix it.
5. Figure 3’s caption, “Source: Nvivo”, citation style must be same throughout the complete manuscript.
6. Discussion needs to be enhance and need critical discussion referencing the others published work with reference to the recommendation received through the questionnaires.
7. Conclusion needs to be rewritten and author must write the major takeaway from the research and analysis being conducted. So the recommendations can be adapted by the related farmers and researchers for application and further extension.
I wish authors a great success.
Author Response
- Author must check for unnecessary capitalization and try to avoid it.
This is checked and capitalizations are made only where they are required
- Section 2.3: the citation must be continuous (numbers), how just after 18, it appears 35,37, and 39. Need careful citation arrangement.
The citations are arranged in continuous format and arrangement is revised.
- Section 3.1: last sentence of the first paragraph, “The peer ….. Science direct.”. Elsevier and science direct are distinct?
The required change has been made. Science direct comes under Elsevier, so to be clear just Elsevier is fine.
- Section 3.2: first paragraph on page 6, the citation style must be uniform throughout the manuscript. As the author changed here to “(Stewart & Shamdasani, 1990)”, fix it.
This is fixed. Replaced intext with cite no.
- Figure 3’s caption, “Source: Nvivo”, citation style must be same throughout the complete manuscript.
This is fixed with appropriate citation.
- Discussion needs to be enhance and need critical discussion referencing the others published work with reference to the recommendation received through the questionnaires.
The discussion section is now enhanced with significance of findings and referencing to other published work with similar results.
- Conclusion needs to be rewritten and author must write the major takeaway from the research and analysis being conducted. So, the recommendations can be adapted by the related farmers and researchers for application and further extension.
The conclusions are rewritten including major takeaways and their significance relating to design and adoption of cost & energy efficient IoT systems for adoption of Precision agriculture practices.

Reviewer 2 Report
In this study, the authors focused on discovering and understanding the decision-making variables related to obstacles with eighteen (n=18) subject experts in IoT-based precision agriculture applications. The following points would clarify my apprehension about this article.
The introduction and related work needs a major rewrite. The introduction needs to be more comprehensive.
The results and scenarios are limited. The punctuation marks throughout the article need to be corrected. Commas, spaces, duplicate commas, etc.
The related work needs to be more comprehensive. (such as 10.1016/j.compeleceng.2021.106982 etc..)
Why is the term “Internet of Things wireless sensor-based” used in the title? As you know If WSN is used, it means IoT is used. It needs to be elaborated/justified if used.
In addition, need to provide performance benchmarking with existing methods and also present the comparison study with existing methods.
Author Response
- The introduction and related work needs a major rewrite. The introduction needs to be more comprehensive.
Introduction section is revised with more comprehension on research problem background and section-wise content of the paper.
- The results and scenarios are limited. The punctuation marks throughout the article need to be corrected. Commas, spaces, duplicate commas, etc.
The limitations of results and scenarios is explained in context of barriers explored (Operational, Technical, Cost and Data management) for adoption of IoT based Precision agriculture practices. The commas and spaces are edited.
- The related work needs to be more comprehensive. (such as 10.1016/j.compeleceng.2021.106982 etc..)
The related work is comprehensive in context of defining research framework for the study. Authors need more explanation on what the reviewer meant by comprehensive.
- Why is the term “Internet of Things wireless sensor-based” used in the title? As you know If WSN is used, it means IoT is used. It needs to be elaborated/justified if used.
Authors understand and edited the title. For the sake of simplicity as this study is based on wireless sensors for precision agriculture applications. The term wireless sensors network (WSN) is a subset of IoT. Therefore, to be explicit the term IoT WSN was used.
- In addition, need to provide performance benchmarking with existing methods and also present the comparison study with existing methods
Authors edited the discussion and conclusion section to provide rigor and significance of results. The references to relevant previous studies have been made to support the findings.

Reviewer 3 Report
Authors proposed a method to adopt IoT-based sensor network technology for smart farm area. In order to investigate their method, authors firstly suggest its motivation and listed some barriers with explanations to adopt the IoT technology. Finally, its feasibility was verified with group interview session. The expression looks like revised version and is fine for me and easy to follow the paper without losing interest. Also, the paper is very well organized. I suggest this paper is ready to be published as it is.
Author Response
Authors appreciate the review. Thanks!